# Analysis of the Phenolic Profile of *Chelidonium majus* L. and Its Combination with Sericin: Balancing Antimicrobial Activity and Cytocompatibility

**DOI:** 10.3390/ijms26209911

**Published:** 2025-10-11

**Authors:** Ana Borges, José Luis Ordóñez-Díaz, Yara Aquino, José Manuel Moreno-Rojas, María Luisa Martín Calvo, Josiana A. Vaz, Ricardo C. Calhelha

**Affiliations:** 1CIMO, LA SusTEC, Instituto Politécnico de Bragança, Campus de Santa Apolónia, 5300-253 Bragança, Portugal; ana.borges@ipb.pt (A.B.); yara.aquino@ipb.pt (Y.A.); calhelha@ipb.pt (R.C.C.); 2Grupo de Investigación en Desarrollo y Evaluación de Formas Farmacéuticas y Sistemas de Liberación Controlada, Facultad de Farmacia, Universidad de Salamanca, Campus Miguel de Unamuno s/n, 37007 Salamanca, Spain; 3Department of Agroindustry and Food Quality, Andalusian Institute of Agricultural and Fisheries Research and Training (IFAPA) Alameda del Obispo, Avda. Menéndez–Pidal s/n., 14004 Córdoba, Spain; josem.moreno.rojas@juntadeandalucia.es; 4Grupo de Investigación en Fisiología y Farmacología, Facultad de Farmacia, Universidad de Salamanca, Campus Miguel de Unamuno s/n, 37007 Salamanca, Spain; marisam@usal.es; 5Research Center for Active Living and Wellbeing (LiveWell), Instituto Politécnico de Bragança, Campus de Santa Apolónia, 5300-253 Bragança, Portugal; josiana@ipb.pt

**Keywords:** sustainable healthcare, biomaterial, silk, greater celandine, phenolic compounds, HPLC-MS/MS, antimicrobial, cytocompatibility

## Abstract

The incorporation of bioactive natural compounds into biomedical applications offers a promising route to enhance therapeutic efficacy while supporting sustainability. In this study, we investigated the synergistic potential of Sericin, a silk-derived biopolymer, and *Chelidonium majus* L. (*C. majus*), a medicinal plant with a diverse phenolic profile, in relation to biological activities relevant for wound care and infection control. A combined experimental strategy was applied, integrating detailed chemical characterization of *C. majus* extracts with antimicrobial and cytocompatibility assays across different Sericin–plant extract ratios (1:1, 1:2, 2:2, and 2:1). Phytochemical analysis identified and quantified 57 phenolic compounds, including high levels of flavonoids (quercetin, kaempferol, isorhamnetin) and phenolic acids (caffeic and ferulic acid). Salicylic acid (123.6 µg/g), feruloyltyramine (111.8 µg/g), and pinocembrin (98.4 µg/g) were particularly abundant, compounds previously reported to disrupt microbial membranes and impair bacterial viability. These metabolites correlated with the strong antimicrobial activity of *C. majus* against Gram-positive strains (MIC = 5–10 mg/mL). In combination with Sericin, antimicrobial performance was ratio-dependent, with higher proportions of *C. majus* (2:1) retaining partial inhibitory effects. Cytocompatibility assays with HFF1 fibroblasts demonstrated low antiproliferative activity across most formulations (GI_50_ > 400 µg/mL), supporting their potential safety in topical applications. Collectively, the results indicate a concentration-dependent interaction between *C. majus* phenolics and the Sericin protein matrix, reinforcing their suitability as candidates for natural-based wound healing materials. Importantly, the valorization of Sericin, an underutilized byproduct of the silk industry, together with a widely accessible medicinal plant, underscores the ecological and economic sustainability of this approach. Overall, this work supports the exploration of the development of biomaterials with potential for advancing tissue repair and wound management.

## 1. Introduction

Plant-derived phenolic compounds have recently gained attention as promising components of wound biomaterials due to their dual ability to exert antimicrobial effects and modulate the redox microenvironment of healing tissues [1,2,3]. Phenolics such as flavonoids and hydroxycinnamic acids can disrupt microbial membranes, inhibit quorum sensing, and interfere with essential enzymatic pathways, thereby offering broad-spectrum antimicrobial protection. Simultaneously, their redox activity plays a critical role in tissue regeneration: by scavenging excess reactive oxygen species (ROS), they protect fibroblasts from oxidative damage, while moderate redox signaling promotes angiogenesis, fibroblast proliferation, and extracellular matrix deposition. Nonetheless, phenolic-rich formulations must be carefully balanced, as elevated concentrations may impair fibroblast viability and compromise tissue repair [3,4,5,6]. These insights underscore the need to evaluate antimicrobial activity and cellular responses in parallel, since the therapeutic value of phenolic-based biomaterials ultimately depends on their capacity to control infection without hindering fibroblast proliferation—an essential event in wound healing [2,3,4].

Recent scaffold-based studies have translated these principles into engineered wound matrices. Polyphenol-infused hydrogels and polymer–polyphenol scaffolds have demonstrated the ability to reduce oxidative stress, sustain antibacterial release, and support re-epithelialization and collagen deposition in both full-thickness and diabetic wound models, while maintaining the viability of fibroblasts [2,7]. Reviews further emphasize that scaffold performance depends strongly on design parameters—including polyphenol type and loading, crosslinking strategy, and release kinetics—which collectively shape the redox dynamics and biological outcomes of wound healing [3,8].

Beyond their general antioxidant properties, mechanistic studies highlight the nuanced role of phenolic compounds in modulating ROS-mediated signaling. While excessive ROS are cytotoxic, controlled ROS levels act as secondary messengers that regulate angiogenesis, fibroblast migration, and extracellular matrix remodeling. Plant-derived phenolics help fine-tune this balance by simultaneously scavenging harmful ROS and supporting physiological redox signaling. A recent review of oxidative stress–driven therapeutic strategies reinforces the relevance of this dual activity in regenerative medicine [9]. Complementary experimental studies show that phenolic-enriched hydrogels enhance vascular endothelial growth factor (VEGF) expression and promote fibroblast proliferation through ROS-mediated activation of MAPK and PI3K/AKT pathways, underscoring their role as bioactive modulators in wound biomaterials. Collectively, these findings demonstrate that phenolics are not passive antioxidants but active regulators of redox biology, with direct implications for scaffold design and wound repair [10,11,12,13,14].

*Chelidonium majus* L. (Greater Celandine) is a medicinal herb widely distributed across Europe and Asia, with documented use extending into North America [15,16,17]. Traditionally used in folk medicine and modern phytotherapy, this perennial plant is valued for its diverse pharmacological activities, which stem from its complex secondary metabolite profile [16,17,18,19,20]. Its phytochemical composition includes isoquinoline alkaloids, flavonoids, saponins, vitamins (notably A and C), mineral elements, sterols, and various acid derivatives [16,17,19,20,21,22]. These compounds contribute synergistically to its antimicrobial, anti-inflammatory, anticancer, immunomodulatory, hepatoprotective, and neuroprotective activities [16,17,18,19,20]. Nevertheless, the variability in phytochemical composition due to environmental factors and the need for clinical validation highlight the importance of continued research [15,16,22].

Among the secondary metabolites of *C. majus*, phenolic compounds are particularly significant [16,17,18,22,23]. These compounds, characterized by aromatic rings with hydroxyl substituents, play vital roles in plant defense and offer well-documented benefits in human health. Their potent antioxidant properties mitigate oxidative stress, while their anti-inflammatory effects are associated with modulation of inflammatory signaling pathways, leading to reduced production of pro-inflammatory cytokines in a concentration- and context-dependent manner. Moreover, phenolics display antimicrobial activity, influence gut microbiota, and exert anticancer effects through apoptosis induction and proliferation inhibition [16,20,21,22,23]. Despite these benefits, important challenges remain. Limited bioavailability and rapid metabolism restrict therapeutic translation, while safety concerns, particularly the reported hepatotoxicity of *C. majus*, underscore the need for rigorous toxicological evaluation. In addition, most evidence derives from in vitro and preclinical studies, with relatively few clinical investigations available. Addressing these gaps is essential to fully establish the safety and efficacy of *C. majus*–derived phenolics for human applications [16,17,22,23].

An emerging strategy involves combining phenolic-rich extracts with Sericin, a natural silk-derived protein renowned for its biocompatibility, antioxidant activity, and film-forming properties. Sericin has found applications in pharmaceuticals, cosmetics, and food technology, and its incorporation into *C. majus* formulations offers a means to improve bioavailability and stability of phenolic compounds. By enhancing solubility and gastrointestinal absorption, Sericin can potentiate the therapeutic efficacy of *C. majus*. Additionally, its film-forming ability supports topical applications, improving local delivery and wound healing outcomes, while encapsulation strategies may enable controlled and targeted release [24,25,26].

The combination of *C. majus* and Sericin thus represents a promising biocompatible approach with applications ranging from pharmaceutical wound formulations to functional foods [16,18,21,22]. Their potential synergistic antioxidant and anti-inflammatory properties could yield innovative, sustainable products to manage oxidative stress and inflammation [24,25,26]. Leveraging Sericin, an underutilized byproduct of the silk industry, together with *C. majus*, a widely available medicinal plant, emphasizes both ecological and economic sustainability [15,16,17,22].

This study aims to develop novel wound-healing formulations enriched with *C. majus* extracts and Sericin through an integrated approach. By characterizing the phenolic composition of *C. majus* and exploring its interactions with Sericin, we assessed antimicrobial properties and cytocompatibility across different ratios. These formulations hold potential as sustainable wound therapeutics, maximizing the bioactive potential of natural compounds while aligning with public health and environmental goals (Figure 1).

## 2. Results and Discussion

The investigation of natural extracts has attracted considerable attention due to their biodegradability, safety, and potential to reduce environmental impact [24,25,27]. In this study, we examined the biological effects of Sericin, a protein derived from *Bombyx mori* cocoons, and *Chelidonium majus* L., a medicinal plant with a long history of therapeutic use. Their antimicrobial activity and cytocompatibility were assessed to evaluate their suitability for biomedical applications. Given the increasing interest in phenolic compounds as key contributors to the pharmacological properties of medicinal plants [17,24,28], the phenolic profile of *C. majus* was also characterized in detail. By combining chemical and biological analyses, this work provides new insights into the potential of *C. majus* and Sericin as natural resources for the development of sustainable wound-healing biomaterials.

### 2.1. Chemical Characterization

Plants synthesize a wide range of chemical compounds broadly classified into primary and secondary metabolites. Primary metabolites—including carbohydrates, lipids, and nucleic acids—are essential for growth, development, and cellular function, providing energy and structural integrity [19]. In contrast, secondary metabolites play specialized roles in plant defense, adaptation, and environmental interaction. These include alkaloids, flavonoids, phenolic acids, terpenoids, and saponins, many of which exhibit considerable pharmacological potential [19,20].

Among these, phenolic compounds are particularly relevant due to their strong antioxidant, antimicrobial, and anti-inflammatory properties, which underpin their value in both medicinal and industrial applications [20]. Sericin, a silk-derived protein, also represents a bioactive compound of significant interest. A recent study demonstrated that the amino acid composition and bioactivity of Sericin are strongly influenced by cocoon origin and extraction methodology [29]. Extracted Sericin contained a wider diversity of amino acids compared to commercial sources, with hydrolyzed forms presenting up to 16 distinct amino acids. Given this comprehensive characterization, the present study did not reassess Sericin’s biochemical composition but instead used these findings as a reference for its biological potential. Importantly, highlighted that high-temperature and high-pressure extraction conditions markedly alter Sericin’s biochemical profile, thereby impacting its suitability for specific biomedical applications [29].

In contrast, the phytochemical profile of *Chelidonium majus* L. (*C. majus*) required further characterization, particularly with respect to its phenolic constituents. While previous research has focused largely on its isoquinoline alkaloids, emerging evidence indicates that flavonoids (e.g., quercetin, kaempferol) and phenolic acids (e.g., caffeic acid, chlorogenic acid) contribute significantly to its antimicrobial and cytocompatibility properties. These compounds are known to neutralize free radicals, alleviate oxidative stress, and modulate inflammatory pathways, reinforcing their therapeutic relevance [19,20,21,22,23].

Given the increasing interest in medicinal plants, the characterization of *C. majus* cultivated in Portugal is particularly relevant. Such efforts support the validation of its traditional uses, ensure quality control in herbal product development, and provide a basis for risk assessment—especially considering the potential toxicity of certain alkaloids, which necessitates controlled dosages. While *C. majus* holds considerable pharmacological promise, detailed phytochemical profiling remains essential to optimize its therapeutic benefits while minimizing risks, thereby ensuring its safe integration into modern healthcare [19,22].

The chemical composition of *C. majus*, however, is highly variable and influenced by factors such as geographical origin, environmental conditions, and extraction techniques. This variability underscores the need for rigorous characterization to standardize and optimize the recovery of bioactive constituents [18,22]. Notably, water-based extraction methods have proven particularly effective for isolating phenolic compounds, supporting both their bioavailability and medicinal efficacy.

#### Phenolic Compounds Profile in *C. majus* 

The phenolic composition of *Chelidonium majus* L. was comprehensively characterized, with 57 compounds tentatively identified and quantified (Table 1). In this sense, 31 of 57 compounds identified were confirmed by analytical standards, being the rest of the compounds tentatively annotated based on MS/MS fragmentation patterns and database comparisons. The concentration of these compounds was semi-quantified using the standard curves of compounds with a similar chemical structure. To confirm these compounds, the availability of analytical standards or the use of complementary techniques such as nuclear magnetic resonance (NMR) would be necessary (Appendix A).

Phenolic acids and flavonoids were the predominant classes. Among the hydroxycinnamic acids, seven derivatives were detected, including dihydrocaffeic acid, caffeic acid, homovanillic acid, *p*-coumaric acid, ferulic acid, 4,5-dicaffeoylquinic acid, and neochlorogenic acid. Six hydroxycinnamic acid amides were also identified, such as *p*-coumaroyltyramine, *N*-caffeoyltyramine, feruloyltyramine, *p*-coumaroylputrescine, feruloylagmatine, and *p*-coumaroyltryptamine. In addition, nine flavones (e.g., apigenin, luteolin, scutellarein, diosmetin, vitexin) and fifteen flavonols (e.g., kaempferol, quercetin, isorhamnetin and their glycosylated derivatives) were detected. These results are consistent with previous phytochemical studies [19,22,23,28,30], reinforcing that *C. majus* is particularly rich in flavonoids and phenolic acids, especially kaempferol, quercetin, and their glycosides. The predominance of these compounds supports the well-documented pharmacological relevance of the species, as quercetin exhibits potent antioxidant and anti-inflammatory effects, kaempferol is associated with anticancer activity, and caffeic and ferulic acids contribute to oxidative stress modulation [16,19,21,23,28].

Quantitative analysis revealed a chemically diverse profile with several metabolites present at particularly high concentrations. Notably, phenylacetic acid (5.774 mg/g), pinocembrin (2.599 mg/g), feruloyltyramine (1.770 mg/g), and salicylic acid (1.545 mg/g) were among the most abundant. Additional compounds such as protocatechuic acid (0.468 mg/g), veratric acid (0.525 mg/g), 4-hydroxyphenylacetic acid (0.659 mg/g), and rutin (0.429 mg/g) were also detected at substantial levels, further enriching the plant’s bioactive profile. In contrast, metabolites such as *p*-coumaroylputrescine, vitexin, and reynoutrin were present at trace concentrations (0.001 mg/g). Hydroxybenzoic acids emerged as the most concentrated group, known for their potent antioxidant, anti-inflammatory, and antimicrobial activities, thereby reinforcing the pharmacological relevance of *C. majus*.

Among the quantified flavonols, isorhamnetin was particularly abundant. As a 3′-O-methylated derivative of quercetin, isorhamnetin is recognized for its antimicrobial, anti-inflammatory, and wound-healing properties. Mechanistically, it exerts antioxidant and cytoprotective effects through the modulation of oxidative stress pathways, while promoting fibroblast migration and angiogenesis, two processes critical for tissue repair. Recent findings also highlight its ability to regulate redox-sensitive signaling cascades, including PI3K/Akt and MAPK pathways, thereby linking its high concentration in *C. majus* extracts to potential wound-healing applications [31].

When compared with previous reports, which described total flavonoid and phenolic acid contents of 137.43 mg/g and 23.67 mg/g, respectively [28], and in different parts of the plant [19,23], the present study provides a more detailed compound-specific profile of the whole plant. These findings are consistent with other studies showing that flavonoids can constitute up to 96% of total phenolics in hydroethanolic extracts of *C. majus* [30], with quercetin-3-O-rutinoside, kaempferol, and isorhamnetin consistently identified as key constituents [28].

From a structure–activity perspective, the predominance of flavonoids and hydroxycinnamic derivatives is noteworthy, as these compounds are characterized by conjugated aromatic systems and hydroxyl substituents known to support antioxidant and antimicrobial activities [32,33]. In particular, phenolic amides such as feruloyltyramine have been reported to disrupt microbial cell integrity, while hydroxybenzoic acids play a central role in mitigating oxidative stress and modulating inflammatory responses [32,34]. The chemical richness of *C. majus* therefore provides a rational basis for evaluating its biological potential.

However, several challenges must be considered before translating these findings into practical applications. Phenolic compounds are inherently prone to oxidative degradation and may undergo rapid metabolism, raising concerns about stability and bioavailability [32,35]. Moreover, while phenolics are generally associated with beneficial effects, *C. majus* is also known to contain alkaloids linked to hepatotoxicity, emphasizing the need for careful dose optimization and safety assessment [36,37]. Another critical limitation is the gap between in vitro characterization and in vivo or clinical evidence; most available data, including the present work, remain at the preclinical stage, limiting conclusions about therapeutic applicability in humans.

These considerations highlight that the chemical composition of *C. majus* not only supports its traditional use but also demands careful attention to compound interactions, stability, and dosing.

A crucial aspect of this work is the consideration of how phenolic bioavailability may be influenced by interactions with Sericin. Previous studies on Sericin [25,29] have shown that its amino acid composition is dominated by serine, aspartic acid, and glycine, residues that promote hydrogen bonding and electrostatic interactions. These structural characteristics suggest that Sericin can engage with polyphenolic compounds primarily through such interactions, potentially modulating their solubility, stability, and release kinetics. Consequently, these protein–polyphenol interactions may play a critical role in determining the biological availability of phenolics in downstream formulations [16,19,23,25,29].

Altogether, the phytochemical richness of *C. majus*, together with the distinctive amino acid profile of Sericin, creates a complementary chemical environment in which protein–polyphenol interactions may occur. Such interactions could potentially influence the solubility, structural integrity, or release profile of phenolic compounds, thereby affecting their availability in downstream biological assays. This framework provides a rationale for subsequent evaluations of antimicrobial activity and cytocompatibility and supports the exploration of multifunctional biomaterials for wound-healing applications, while acknowledging that the extent and functional consequences of these interactions remain to be experimentally determined.

**Table 1 ijms-26-09911-t001:** Tentative identification and quantification by LC-HRMS/MS of phenolic compounds in *Chelidonium majus* L.

Group	Phenolic Compounds	Molecular Formula	Retention Time	Calculated	Experimental	ms/ms				Error	MSIMI Level ^a^	Mean	SD	Reference
			(min)	[m/z]-	[m/z]-					(ppm)		(mg/g)	(mg/g)	
**Phenolic Acids**														
Hydroxybenzoic acids	Protocatechuic acid	C_7_H_6_O_4_	5.8	153.0193	153.0192	55.6681	81.0347	108.0216	109.0295	6.76	2	0.468	0.077	[38]
	Veratric Acid	C_9_H_10_O_4_	6.56	181.0506	181.0508	92.92	107.0502	136.91	163.0399	1.33	2	0.525	0.077	
	Salicylic acid	C_7_H_6_O_3_	7.21	137.0244	137.0244	65.0397	93.0346	96.0093	108. 8996	8.17	2	1.545	0.22	
	4-H-phenylacetic acid	C_8_H_8_O_3_	7.52	151.04	151.0399	44.9984	94.9573	107.0503	122.952	6.68	1	0.659	0.083	Standard
	Vanillic acid	C_8_H_8_O_4_	7.74	167.0349	167.0353	108.0217	123.0448	138.929	152.0117	8.53	1	0.153	0.02	
	Ellagic Acid	C_14_H_6_O_8_	8.33	300.9989	300.999	126.8811	257.2135	265.1811	283.1914	3.94	2	0.235	0.035	[39]
Hydroxycinnamic acids	Dihydrocaffeic acid	C_9_H_10_O_4_	6.95	181.0506	181.0508	59.6585	92.92	136.91	181.0509	1.33	1	0.174	0.002	Standard
	Neochlorogenic acid	C_16_H_18_O_9_	7.01	353.0878	353.0871	106.0289	133.0151	178.0739	310.1153	1.27	2	0.192	0	[40]
	Caffeic acid	C_9_H_8_O_4_	7.57	179.0349	179.0354	44.9982	90.9243	107.0504	135.0452	8.9	1	0.04	0.005	Standard
	Homovanillic acid	C_9_H_10_O_4_	8.28	181.0506	181.0508	92.92	122.8946	136.9099	152.9176	7.2	1	0.006	0.001	Standard
	*p*-Coumaric acid	C_9_H_8_O_3_	8.56	163.04	163.0401	93.0344	119.0503	121.0522	162.8392	1.15	2	0.058	0.008	[38]
	Ferulic acid	C_10_H_10_O_4_	8.8	193.0506	193.05	121.0659	134.0374	149.0972	178.0273	6.24	1	0.055	0.01	Standard
	4,5-Dicaffeoylquinic acid	C_25_H_24_O_12_	9.01	515.1194	515.1195	93.0345	135.0452	173.0455	191.0563	2.15	1	0.178	0.004	
Hydroxycinnamic acid amides	Feruloylagmatine	C_15_H_22_N_4_O_3_	8.08	305.1619	305.1589	97.0659	135.0816	249.1497	287.1646	−6.05	2	0.006	0.001	[41]
	*p*-Coumaroylputrescine	C_13_H_18_N_2_O_2_	9.23	233.1295	233.1659	120.0819	164.9275	215.1557		4.93	2	0.001	0	[42]
	N-Caffeoyltyramine	C_17_H_17_NO_4_	9.29	298.1084	298.1086	75.0088	135.03	179.0357	206.9749	4.31	2	0.008	0	[43]
	*p*-Coumaroyltyramine	C_17_H_17_NO_3_	9.96	282.1135	282.1136	119.0503	132.0579	145.0298	162.0561	4.07	2	0.901	0.002	[44]
	Feruloyltyramine	C_18_H_19_O_4_N	10.16	312.1241	312.12418	148.0531	178.0511	190.0513	297.101	3.67	2	1.77	0.242	[45]
	*p*-Coumaroyltryptamine	C_19_H_18_N_2_O_2_	11.8	305.1295	305.1759	135.0816	249.1497	287.1646		8.41	2	0.002	0	[46]
**Flavonoids**														
Flavanol	Catechin	C_15_H_14_O_6_	6.98	289.0717	289.0718	146.9387	162.8392	190.9286	197.8081	4.17	1	0.002	0	Standard
	Epicatechin	C_15_H_14_O_6_	7.12	289.0717	289.0718	160.8422	181.0508	190.9287	195.8111	4.17	1	0.021	0	
Flavanone	Hesperidin	C_28_H_34_O_15_	9.01	609.1824	609.1462	151.0033	243.0298	271.0244	300.0277	2.01	1	0.181	0.024	
	Naringenin	C_15_H_12_O_5_	11.32	271.0611	271.0613	59.0139	62.5824	95.9463	198.829	4.68	1	0.59	0.031	
	Hesperetin	C_16_H_14_O_6_	11.57	301.0717	301.2022	126.8811	221.1912	265.1811	283.1914	4.18	1	0.038	0.004	
	Pinocembrin	C_15_H_12_O_4_	13.62	255.0662	255.0664	151.0037	171.0455	213.0554		4.8	2	2.599	0.094	[47]
Flavone	Vitexin	C_21_H_20_O_10_	8.28	431.0983	477.10413	1101.0243	114.6059	205.9066	348.5617	8.28	1	0.001	0	Standard
	Cynaroside	C_21_H_20_O_11_	8.44	447.0932	447.0929	227.0347	255.0298	284.0329	285.0406	0.8	1	0.002	0	
	Diosmin	C_28_H_32_O_15_	8.87	607.1668	607.1673	277.0348	283.0256	299.0564		2.65	1	0.024	0.005	
	Apigetrin	C_21_H_20_O_10_	9.01	431.0983	431.0983	151.0038	211.0406	268.038	311.0568	1.08	1	0.002	0	
	Diosmetin-7-glucoside	C_22_H_22_O_11_	9.18	461.1089	461.1087	63.0241	255.0298	284.0328	299.0564	2	1	0.023	0.001	
	Luteolin/Scutellarein	C_15_H_10_O_6_	10.38	285.0404	285.0405	121.0296	126.8811	133.0295	136.9099	4.08	1	0.003	0	
	Apigenin	C_15_H_10_O_5_	11.23	269.0455	269.0455	57.8443	64.8013	65.6261		3.9	1	0.004	0	
	Diosmetin	C_16_H_12_O_6_	11.4	299.0561	299.0561	59.014	69.6641	164.8297	255.3027	1.1	1	0.002	0	
Flavonol	Isorhamnetin-rutinoside- glucoside	C_34_H_42_O_21_	7.15	785.2145	785.2154	315.051	623.1617			2.5	2	0.028	0	[30]
	Quercetin-3-rhamnosylrutinoside	C_33_H_40_O_20_	7.66	755.204	755.2044	300.0276	271.025	255.0302	243.0297	1.49	2	0.141	0.003	[28]
	Isorhamnetin rutinoside-rhamnoside	C_34_H_42_O_20_	7.96	769.2196	769.2206	315.051	299.0197	271.0249	243.0299	2.69	2	0.082	0.001	[30]
	Rutin	C_27_H_30_O_16_	8.12	609.1461	609.1466	243.0296	271.0246	300.0276	405.7915	2.72	1	0.429	0.035	Standard
	Quercetin-glucoside	C_21_H_20_O_12_	8.4	463.0881	463.0878	151.0037	255.0303	271.0245	300.0275	1.63	2	0.021	0.001	[28]
	Kaempferol 3-rutinoside	C_27_H_30_O_15_	8.53	593.1511	593.1517	183.0449	227.035	255.0301	285.0406	2.85	1	0.069	0.011	Standard
	Isorhamnetin rutinoside	C_28_H_32_O_16_	8.58	623.1617	623.1619	315.051	299.0197	271.0249	243.0299	2.03	2	0.32	0.024	[30]
	Guaiaverin/Reynoutrin	C_20_H_18_O_11_	8.73	433.0776	433.0764	151.0044	255.0307	271.025	300.0276	−0.118	1	0.001	0	Standard
	Quercitrin/Quercetin 3-rhamnoside	C_21_H_20_O_11_	8.84	447.0932	447.0929	183.0463	255.0295	284.0329	300.0275	1.79	1	0.025	0.001	
	Isorhamnetin-3-glucoside	C_22_H_22_O_12_	8.89	477.1038	477.1041	199.0409	243.0298	285.0407	314.0435	2.88	1	0.005	0	
	Quercetin	C_15_H_10_O_7_	10.44	301.0353	301.2022	126.8811	221.1912	265.1811	283.1914	4.18	1	0.011	0	
	Kaempferol	C_15_H_10_O_6_	11.37	285.0404	285.2075	52.2938	126.8809	267.1924		5.3	1	0.007	0	
	Isorhamnetin	C_16_H_12_O_7_	11.53	315.051	315.0512	288.9368	294.8804	310.858	313.0358	1.34	1	0.042	0	
**Coumarins**														
	Esculetin/Aesculetin	C_9_H_6_O_4_	7.57	177.0193	177.0193	89.0393	105.0345	133.0296	149.0244	6.18	2	0.061	0.012	[48]
	Umbelliferone	C_9_H_6_O_3_	7.68	161.0244	161.0819	87.0451	115.0401	117.0557	132.9808	6.92	2	0.004	0	[49]
	Scoparone	C_11_H_10_O_4_	8.78	205.0506	205.0506	125.8733	157.8631	160.8422	161.8501	1.07	2	0.004	0	[50]
	Scopoletin	C_10_H_8_O_4_	8.98	191.0349	191.0193	102.9488	111.0088	146.9387	176.0115	3.82	2	0.008	0.001	[45]
**Stilbenes**														
	Resveratrol	C_14_H_12_O_3_	9.37	227.0713	227.1289	130.9838	165.1285	183.1392	227.1289	1.15	2	0.01	0	[51]
	Piceatannol	C_14_H_12_O_4_	10.98	243.0662	243.1236	146.961	199.1338	174.9564	225.1131	4.07	2	0.01	0	[52]
**Others**														
Phenolic Aldehyde	Syringaldehyde	C_9_H_10_O_4_	6.56	181.0506	181.0508	92.92	136.91	152.9175	181.0509	1.33	2	0.124	0.008	[53]
	Vanillin	C_8_H_8_O_3_	8.56	151.04	151.04	108.0217	122.9521	136.0166	151.0401	7.34	1	0.129	0.01	Standard
	Protocatechualdehyde	C_7_H_6_O_3_	10.3	137.0244	137.0244	92.92	108.9	124.8951		8.53	2	0.186	0.037	[54]
Phenylethanoids	Hydrotyrosol	C_8_H_8_O_3_	8.56	151.04	151.04	108.02	122.9521	136.0166		7.34	1	0.061	0.012	Standard
Phenylpropanoids	Phenylacetic acid	C_8_H_8_O_2_	9.88	135.0451	135.0451	59.0139	87.0088	90.9241		1.09	1	5.774	0.323	

^a^ Metabolite Standards Initiative metabolite identification (MSIMI) levels. Reference compounds were available for all compounds identified at MSIMI level 1. Compounds at MSIMSI level 2 were tentatively identified.

### 2.2. Antimicrobial Activity

The antimicrobial evaluation of *C. majus* and Sericin extracts, alone and in combination, is summarized in Table 2. As expected, *C. majus* alone exhibited measurable antimicrobial activity, with MIC values ranging from 5 to 10 mg/mL across all tested bacteria and fungi and MBC/MIC ratios of 1–2, indicating modest bactericidal effects. In contrast, Sericin alone was inactive, except for the commercial variant (Sigma-Aldrich), which showed activity against *Cutibacterium acnes* and *Staphylococcus epidermidis* (MIC = 5 mg/mL). When combined, the extracts generally demonstrated reduced antimicrobial activity, with the outcome depending primarily on the concentration ratio rather than the Sericin source. Notably, the 2:1 ratio of *C. majus* to Sericin retained partial inhibitory effects against *S. aureus*, *S. epidermidis*, and *C. acnes*, whereas the 1:2 ratio abolished all activity. Combinations containing Sericin S3 consistently produced the weakest outcomes.

The inclusion of reference antibiotics (ampicillin and vancomycin for Gram-positive bacteria) ensured the reliability of the antimicrobial assays and provided a benchmark for interpreting the activity of the extracts. *Candida albicans*, being a yeast species, was tested using fluconazole as the reference antifungal control. The positive controls exhibited strong inhibitory effects against *S. aureus*, *S. epidermidis*, and *P. acnes*, with MIC values within the expected ranges reported for clinical isolates in the recent literature [55,56]. Similarly, fluconazole demonstrated potent inhibitory activity against *C. albicans*, with MIC values also consistent with established benchmarks [56,57]. This concordance corroborates the robustness of the assay conditions and validates the observed antimicrobial performance of the *C. majus* extract. In comparison, the extract demonstrated moderate inhibitory activity (MIC = 5 mg/mL), which, although requiring higher concentrations than conventional antibiotics and antifungals, indicates a reproducible antimicrobial effect attributable to its phenolic profile. Furthermore, the use of these controls confirmed that the reduced activity observed in sericin-rich combinations was not a methodological artifact but rather reflected genuine modulation of extract efficacy through protein–phenolic interactions.

Moreover, these findings are consistent with prior reports showing that aqueous Sericin solutions or Sericin-based hydrogels lack antimicrobial effects against common wound pathogens (*S. aureus*, *Pseudomonas aeruginosa*, *Escherichia coli*) [58]. In contrast, the antimicrobial potential of *C. majus* metabolites has been confirmed in other systems; for example, ref. [59] demonstrated that *C. majus* incorporated into bacterial nanocellulose matrices inhibited both planktonic and biofilm-forming cells, including *S. aureus*, *P. aeruginosa*, and *Candida albicans*.

A possible association can be drawn between the antimicrobial activity of *C. majus* and its phenolic composition. The extract contained high levels of metabolites such as salicylic acid, feruloyltyramine, pinocembrin, and phenylacetic acid, all previously reported to affect microbial integrity or signaling.

Recent studies suggest that these compounds act through complementary mechanisms involving disruption of microbial membrane integrity, interference with enzyme function, and induction of oxidative stress. Salicylic acid and related hydroxybenzoic acids are known to impair cell wall synthesis and alter membrane permeability, whereas flavanones such as pinocembrin can inhibit ATP production and increase membrane depolarization, leading to leakage of intracellular components [Refs]. Feruloyltyramine and phenylacetic acid, on the other hand, have been associated with inhibition of quorum sensing and modulation of oxidative stress–related pathways in bacteria.

However, this study did not directly test the contribution of individual metabolites, and therefore, no causal link can be established. When combined with Sericin, activity decreased in a ratio-dependent manner, suggesting that the presence of Sericin may reduce the bioavailability or effective concentration of phenolic compounds.

One plausible explanation involves phenolic–protein interactions [60,61]. The interaction between Sericin and phenolic compounds may attenuate these effects by reducing the availability of free phenolic hydroxyl groups, which are essential for antimicrobial action. Sericin, rich in polar amino acids (serine, glycine, aspartic acid), can form hydrogen bonds and electrostatic interactions with phenolic structures, thereby stabilizing the compounds but decreasing their reactivity. This interaction likely limits the diffusion of active metabolites across microbial membranes, resulting in lower observed antimicrobial potency [1,62]. Mechanistically, the antimicrobial activity of *C. majus* phenolics has been linked to reactive oxygen species (ROS) generation, protein oxidation, and cell wall disruption—processes that may be partially hindered when Sericin is present as a binding matrix [4,63,64].

Differences among Sericin sources may further reflect variations in amino acid composition, solubility, and molecular weight distribution resulting from distinct extraction methods. Commercial Sericin preparations typically contain smaller, more soluble fractions, which may explain the limited intrinsic activity observed here, whereas crude extracts (S1–S3) produced less consistent outcomes. Such variability is well documented, as extraction protocols strongly influence Sericin’s biochemical profile and functional properties [25,29,65].

Comparisons with other biopolymer–plant systems highlight the broader translational relevance of these results. For example, quercetin nanocrystal–loaded alginate hydrogels exhibited sustained release, antimicrobial efficacy, and accelerated wound closure in vivo [66]. Similarly, gelatin/alginate scaffolds co-loaded with caffeic acid and quercetin achieved multifunctional effects by combining antimicrobial, antioxidant, and pro-healing properties [67]. Injectable photocrosslinked hydrogels incorporating quercetin-loaded metal–organic frameworks were also shown to enhance re-epithelialization while suppressing bacterial growth [68]. These examples demonstrate how the integration of phenolic biopolymers can yield multifunctional wound dressings. Within this framework, the unique behavior of *C. majus*–Sericin combinations underscores the importance of matrix composition and bioactive interactions, highlighting opportunities for rationally optimizing antimicrobial and cytocompatible formulations.

Finally, the microbial strains selected in this study—*S. epidermidis* and *P. acnes*—were chosen for their clinical relevance to skin health and wound healing. *S. epidermidis* is associated with biofilm formation that can delay tissue repair, while *C. acnes* contributes to inflammatory conditions. Evaluating *C. majus*–Sericin combinations against these strains, therefore, provides insight into their potential applicability in dermatological and wound care contexts, even though the results show that Sericin attenuates rather than enhances the antimicrobial effects of *C. majus* under the tested conditions. Although the findings contrast with the initial hypothesis of a synergistic interaction, the attenuation of antimicrobial activity by Sericin offers relevant mechanistic insight. Sericin appears to act as a modulator that stabilizes but also limits the bioavailability of *C. majus* phenolics through protein–phenolic interactions. These results, while representing negative outcomes in terms of antimicrobial potency, are valuable in defining the physicochemical constraints of such systems. They highlight the trade-off between bioactivity and cytocompatibility, providing a useful framework for optimizing future Sericin-based formulations in wound-healing and biomedical applications.

### 2.3. Cytocompatibility Evaluation

The cytocompatibility of the tested samples, expressed as GI_50_ values (µg/mL) in HFF1 fibroblasts, is summarized in Table 3. Most individual extracts and combinations demonstrated high cytocompatibility, with GI_50_ values above 400 µg/mL, indicating negligible cytotoxicity. Several formulations, including 2CS1, 2CS2, 3CS2, 3CS3, 4CS3, and 4CS4, displayed moderate cytocompatibility, with GI_50_ values between 150 and 377 µg/mL. The lowest value was observed for the 2CS1 formulation (2:1 ratio of *C. majus* with Sericin from Castelo Branco; GI_50_ = 150 ± 3.6 µg/mL), suggesting a concentration-dependent effect at higher doses. Importantly, even the lowest GI_50_ values remained within acceptable safety margins, supporting the potential of these natural extracts for biomedical use. These results highlight the importance of optimizing extract ratios to balance biological activity with cellular safety.

The findings are consistent with previous reports showing dose-dependent cytotoxicity of *C. majus* in different systems. A recent study described strong activity in pod and flower extracts, with more moderate inhibition from leaf, stem, and root fractions [19]. By contrast, Sericin has repeatedly been shown to be biocompatible. For example, another study demonstrated that Sericin–chitosan scaffolds at a 1:1 ratio were non-toxic to dermal fibroblasts over eight days [69]. Taken together, the present results suggest that while *C. majus* contributes measurable bioactivity, Sericin may mitigate potential cytotoxic effects, thereby supporting fibroblast viability.

A possible association between the safety profile of *C. majus* and its phytochemistry can be noted. The extract is rich in hydroxybenzoic acids, hydroxycinnamic acids, flavones, and flavonols—compounds widely reported for their antioxidant and anti-inflammatory effects. Metabolites such as salicylic acid, feruloyltyramine, pinocembrin, and phenylacetic acid were present at relatively high levels and may contribute to maintaining cytocompatibility, although no direct causative link was tested here. Hydroxybenzoic acids, in particular, are known to reduce oxidative stress, which may partially explain the generally favorable cytocompatibility observed.

A deeper insight into the bioactivity of the *C. majus*–Sericin formulations can be considered in terms of specific compounds and potential mechanisms. Phenolic constituents such as hydroxybenzoic acids, hydroxycinnamic acids, flavones, and flavonols are likely contributors to both cytoprotective and antimicrobial effects. These compounds can modulate oxidative stress in fibroblasts, supporting cell viability, while simultaneously exhibiting antimicrobial activity through interactions with microbial membranes or enzymes. Sericin may interact with these phenolics via hydrogen bonding or hydrophobic interactions, which could influence their bioavailability and release kinetics, thereby balancing antimicrobial potency with cytocompatibility [70,71,72,73]. Although the precise cellular pathways were not directly assessed in this study, the combination of antioxidant phenolics and Sericin’s protective properties suggests a mechanism whereby fibroblast survival is maintained even in the presence of bioactive compounds with antimicrobial potential.

Variability among Sericin sources may also explain differences in cytocompatibility. Commercial Sericin typically consists of lower molecular weight fractions with higher solubility, properties that can favor biocompatibility. In contrast, crude preparations may retain larger peptides, pigments, or residual impurities that influence cell behavior. Such differences, reported in prior studies [29,65,74], suggest that Sericin’s extraction method strongly shapes its biochemical profile and interactions with *C. majus*, influencing cytocompatibility outcomes.

Comparable biopolymer–phenolic systems help contextualize these findings. For instance, quercetin nanocrystal–loaded alginate hydrogels accelerated wound closure in vivo while maintaining fibroblast viability in vitro [66]. Similarly, gelatin/alginate scaffolds reinforced with TiO_2_ nanoparticles released quercetin, caffeic acid, and allantoin while preserving cytocompatibility [67]. More recently, 3D-printed Alg–Gel scaffolds functionalized with TA@ZIF-8 supported fibroblast proliferation and adhesion alongside antimicrobial effects [75]. In this context, the present results suggest that higher Sericin proportions may help preserve fibroblast viability by moderating phenolic exposure, even if this reduces antimicrobial potency. While higher Sericin ratios were associated with reduced antimicrobial activity in other assays, these formulations consistently maintained excellent cytocompatibility, highlighting a trade-off between antimicrobial potency and cellular safety that may guide optimal formulation design.

Finally, the HFF1 human fibroblast line was chosen because of its relevance to skin physiology and wound healing. These cells contribute to extracellular matrix production, collagen synthesis, and tissue remodeling, making them a widely used model for evaluating cytocompatibility. Their use strengthens the translational value of the present findings, which suggest that *C. majus*–Sericin formulations can achieve acceptable safety levels while maintaining biological activity, supporting further exploration for wound-healing applications.

## 3. Materials and Methods

### 3.1. Chelidonium majus L. (C. majus)

*C. majus* whole plant was purchased in dried form from Natura Store (Setúbal, Portugal; Lot no. 11CELL435J191S), packaged and distributed by Alfredo Augusto Tavares & Sucessores, Lda. According to supplier information, the material originated from Bulgaria. Upon reception, the dried material was stored in a dry, dark environment at room temperature until extraction. Hydroethanolic extracts were obtained using a solid–liquid extraction method. Briefly, 1.5 g of dried plant material was mixed with 30 mL of an ethanol–water solution (80:20, *v*/*v*; providing an intermediate polarity optimal for the extraction of a broad spectrum of phytochemicals, particularly polyphenols and flavonoids) and stirred at 500 rpm with a magnetic stirrer for 1 h at room temperature. The suspension was then filtered through quantitative ashless cellulose filter papers (CHMLAB, Grade F2040, Ref. DA0073A, 125 mm diameter, ash content ≤ 0.01%), and the retained residue was re-extracted under identical conditions. The combined filtrates were concentrated by removing ethanol under reduced pressure using a rotary evaporator (Hei-VAP Core, Heidolph Instruments, Schwabach, Germany) at 40 °C, 120 rpm, and a pressure of approximately 175 mbar [76,77]. The remaining aqueous phase was frozen at −80 °C and lyophilized with a Büchi R-20 lyophilizer (Flawil, Switzerland). The resulting extract was stored in the dark until further analysis.

### 3.2. Sericin

Sericin was extracted from *Bombyx mori* cocoons collected in different regions of Portugal. Cocoons from Bragança (2019) were provided by Casa da Seda (Centro de Ciência Viva de Bragança), and those from Castelo Branco (2019) were obtained from the Associação Portuguesa de Pais e Amigos do Cidadão Deficiente Mental (APPACDM). Commercial Sericin samples were purchased from Sigma-Aldrich and FUJIFILM Wako Chemicals (2023). The extraction procedure followed an adaptation of [29]. Briefly, cocoons were manually cleaned, cut into ~1 cm^2^ fragments, and washed three times with deionized water. The cleaned material was dried at 50 °C, weighed, and subjected to degumming by immersion in ultrapure water (1.5% *w*/*v*), followed by autoclaving at 120 °C for 30 min. The resulting aqueous Sericin solution was filtered to remove residual fibers, stored in sealed containers, frozen at −80 °C, and lyophilized. The obtained Sericin powders were stored at 4 °C until further use.

### 3.3. Combination of the Extracts

Formulations combining *C. majus* and Sericin extracts were prepared at different concentration ratios (1:1, 1:2, 2:2, and 2:1). Each ratio was designed for the respective assays to evaluate potential synergistic or antagonistic interactions. This systematic approach enabled a comprehensive assessment of their combined effects, providing insights into the interaction dynamics between *C. majus* and Sericin under varying experimental conditions (Table 4).

### 3.4. Identification and Quantification of Phenolic Compounds in C. majus

The tentative identification and quantification of the phenolic compounds in the *C. majus* samples were performed using a micro-flow LC-HRMS/MS mass spectrometer system (Thermo Fischer Scientific, San José, CA, USA) comprising a Vanquish Neo UHPLC system coupled to an Orbitrap Exploris 240 mass spectrometer. The chromatography separation was performed using an Acclaim™ PepMap™ 100 C18 columns (150 mm × 1 mm, 2 μm) (Thermo Fischer Scientific, San José, CA, USA). The mobile phase A was water and mobile phase B was acetonitrile, both acidified with 0.1% of formic acid. The gradient time was 20 min (0–2 min, 0% B; 2–16 min, 0–95% B; 16–20 min, 95% B) at a flow rate of 50 μL/min and sample injection was 0.2 μL. The Exploris 240 MS was operated in full scan in negative mode (*m*/*z* 100–1500 Da) and tandem MS/MS using data-dependent mass scan mode. The resolution was fixed at 180,000 and 15,000 to full scan and date-dependent mass scan, respectively. The parameters of Heated-Elestrospray Ionization (HESI) were the following: spray voltage (2500 V), sheath gas (20 units), auxiliary gas (5 units), ion transfer tube temperature (325 °C), vaporizer temperature (120 °C), and RF lens (70%). Data acquisition and analysis were performed using Trace Finder 5.1 software and Xcalibur 4.6 software (Thermo Scientific, San José, CA, USA). The identification of phenolic compounds was performed by comparing the exact mass as well as the retention time with available standards. When the standards were not available, the phenolic compounds were tentatively identified by comparing the theoretical exact mass and experimental accurate mass of the molecular ion, these were checked with chemical compound databases, such as MzCloud, Phytohub, Phenol Explorer, PubChem, etc., with an error tolerance ≤5 ppm. The quantification of phenolic compounds was performed using standard curves of reference standards, in the absence of these, the quantification was carried out using a closely related parent compound.

### 3.5. Bioactivities Evaluation

The antimicrobial and cytocompatibility properties of the different extract combinations were evaluated due to their direct relevance to critical phases of the wound healing process, including infection control, modulation of oxidative stress, and maintenance of tissue compatibility.

#### 3.5.1. Antimicrobial Assay

The antimicrobial assay was performed following the protocol described by [56]. A colorimetric method was used to determine the minimum inhibitory concentration (MIC) of the samples. Extracts were dissolved in DMSO/MHB, serially diluted, and transferred into microplates containing MHB/TSB, the test sample, and the microbial inoculum. MIC values were determined by adding 0.2 mg/mL *p*-iodonitrotetrazolium chloride, followed by incubation at 37 °C for 30 min. The minimum bactericidal concentration (MBC) was established by plating aliquots from wells showing no visible growth. The antimicrobial potential was tested against four clinically relevant skin-associated strains: three Gram-positive bacteria—*Staphylococcus aureus*, *Staphylococcus epidermidis*, and *Propionibacterium acnes*—as well as the fungus *Candida albicans*. All strains were isolated from patients hospitalized in different departments of the Hospital Center of Trás-os-Montes and Alto Douro (Vila Real, Portugal). Appropriate controls were included: (i) MHB/TSB only, (ii) extract only, and (iii) medium with antibiotic as negative controls, and (iv) MHB with inoculum as a positive control. Ampicillin and vancomycin served as reference antibiotics for Gram-positive bacteria. Results were expressed as MIC, minimum bactericidal concentration (MBC), or minimum fungicidal concentration (MFC), in mg/mL.

#### 3.5.2. Cytocompatibility Assay

The cytocompatibility of the extracts was evaluated using the sulforhodamine B (SRB) colorimetric assay, following the protocol described by [78]. Human fibroblast cells (HFF1), obtained from the Leibniz Institute DSMZ—German Collection of Microorganisms and Cell Cultures GmbH (Braunschweig, Germany), were used as the in vitro model. Cytocompatibility was quantified based on the GI_50_ value, defined as the concentration required to inhibit 50% of cell proliferation. Extracts were classified according to the criteria established by the U.S. National Cancer Institute: low cytocompatibility (GI_50_ ≤ 20 µg/mL), moderate (GI_50_ = 21–200 µg/mL), high (GI_50_ = 201–400 µg/mL), and very high (GI_50_ > 400 µg/mL). A cell suspension without extract served as the control. Results are expressed as GI_50_ values (µg/mL), with higher values indicating greater cytocompatibility.

### 3.6. Statistical Analysis

Each assay was conducted in duplicate, and the results were expressed as mean values ± standard deviation, calculated using Microsoft Excel (Microsoft Corporation, Redmond, WA, USA).

## 4. Conclusions

Comprehensive chemical profiling of *C. majus* revealed a diverse set of 57 phenolic compounds, with flavonols, flavones, and hydroxycinnamic acids (including amides) as the predominant groups. Among these, salicylic acid, feruloyltyramine, pinocembrin, and phenylacetic acid were particularly abundant, while hydroxybenzoic acids represented the most concentrated class overall. Additional metabolites, such as protocatechuic acid, veratric acid, and rutin, also contributed substantially, whereas *p*-coumaroylputrescine and vitexin were detected only in trace amounts. This chemical richness is consistent with previous reports on the phytochemical diversity of *C. majus* and supports its relevance as a source of bioactive compounds.

Biological assays further clarified the functional significance of these extracts. *C. majus* alone exhibited consistent antimicrobial activity, with MIC values of 5–10 mg/mL against all tested bacterial and fungal strains. In contrast, Sericin alone was inactive, except for the commercial variant, which showed limited effects against *C. acnes* and *S. epidermidis*. When combined, antimicrobial activity was generally attenuated, with the ratio of *C. majus* to Sericin being the primary determinant. Formulations with higher *C. majus* content (2:1) retained partial inhibitory activity, whereas Sericin-rich formulations (1:2) abolished antimicrobial effects. These results suggest that Sericin may reduce the bioavailability or effective concentration of phenolic compounds, potentially through protein–polyphenol interactions, and underscore the concentration-dependent nature of the observed outcomes.

Cytocompatibility testing in HFF1 fibroblasts showed that most extracts and combinations maintained high cell viability (GI_50_ > 400 µg/mL). Several formulations displayed moderate cytocompatibility (GI_50_ = 150–377 µg/mL), with the lowest value recorded for the 2CS1 formulation (GI_50_ = 150 ± 3.6 µg/mL). Although these results indicate a possible concentration-dependent cytotoxic effect, all values remained within acceptable safety thresholds. The data therefore suggest that while *C. majus* provides measurable bioactivity, Sericin may help to balance these effects, contributing to the overall cytocompatibility of the formulations.

Together, these findings indicate that *C. majus*–Sericin combinations exhibit ratio-dependent biological behavior, in which Sericin does not act as a synergistic enhancer of antimicrobial activity but may play a role in moderating cytotoxicity. Similar dynamics have been described in other biopolymer–phenolic systems, where interactions between proteins and polyphenols influence both bioavailability and functional outcomes. Within this framework, careful adjustment of formulation ratios emerges as a critical step for balancing antimicrobial activity and cellular safety. Although these results diverge from the initial expectation of a synergistic interaction between *C. majus* and Sericin, they provide insight into the behavior of such biopolymer–phenolic systems. The observed attenuation of antimicrobial activity upon Sericin incorporation suggests a modulatory rather than potentiating role, likely arising from protein–phenolic associations that limit the bioavailability of active metabolites. Importantly, these findings constitute a valuable result, delineating the functional boundaries of Sericin–phenolic interactions and underscoring the necessity of precise formulation optimization to achieve a balance between antimicrobial efficacy and cytocompatibility.

Beyond their biological properties, these extracts also carry environmental and economic advantages. Sericin, typically discarded as a byproduct of silk production, can be valorized as a biocompatible protein source, while *C. majus*, a widely distributed wild-growing plant, represents a sustainable source of phenolic compounds. Integrating these natural materials into biomedical applications offers the dual benefit of reducing waste and advancing eco-friendly healthcare solutions.

In conclusion, this study demonstrates that *C. majus*–Sericin formulations show concentration-dependent antimicrobial and cytocompatibility outcomes. While Sericin did not enhance the antimicrobial activity of *C. majus*, it may contribute to maintaining biocompatibility, suggesting a modulatory rather than synergistic role. These results highlight the importance of ratio optimization and provide a basis for further investigation into the interactions between Sericin proteins and *C. majus* phenolics. Future work should focus on elucidating the molecular mechanisms underlying these interactions and refining formulations for wound-healing applications, while also considering the sustainability benefits of these natural bioresources.

## Figures and Tables

**Figure 1 ijms-26-09911-f001:**
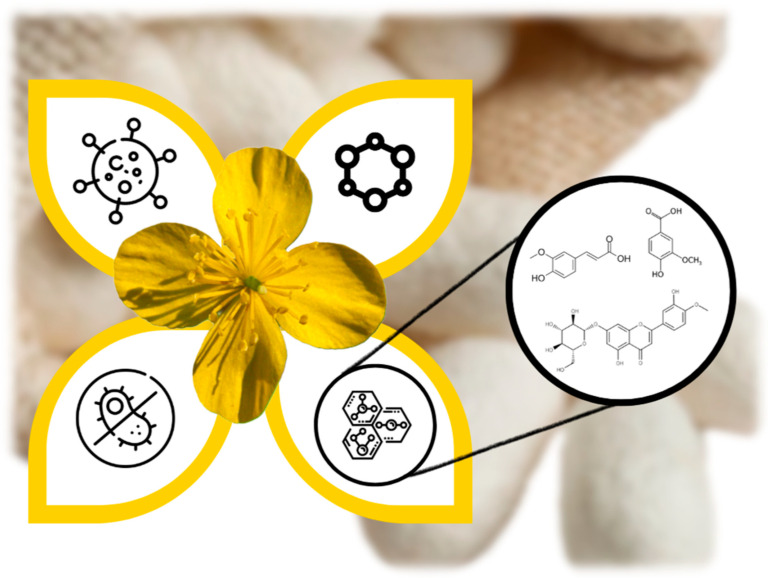
Graphic overview of the study.

**Table 2 ijms-26-09911-t002:** Antimicrobial potential of the different samples studied.

			Antibacterial Activity						Antifungal Activity	
			Gram-Positive Bacteria							
			*Cutibacterium acnes*		*Staphylococcus aureus*		*Staphylococcus epidermidis*		*Candida* *albicans*	
			MIC	MBC	MIC	MBC	MIC	MBC	MIC	MFC
Samples	C	10 mg/mL	5	>10	5	10	10	>10	10	>10
	S1	10 mg/mL	>10	>10	>10	>10	>10	>10	>10	>10
	S2	10 mg/mL	10	>10	>10	>10	>10	>10	>10	>10
	S3	10 mg/mL	5	>10	>10	>10	5	>10	>10	>10
	S4	10 mg/mL	>10	>10	>10	>10	>10	>10	>10	>10
	1CS1	10 mg/mL	>10	>10	10	>10	10	>10	>10	>10
	1CS2	10 mg/mL	10	>10	10	10	>10	>10	>10	>10
	1CS3	10 mg/mL	10	>10	10	>10	10	>10	>10	>10
	1CS4	10 mg/mL	10	>10	>10	>10	10	>10	>10	>10
	2CS1	10 mg/mL	>10	>10	>10	>10	>10	>10	>10	>10
	2CS2	10 mg/mL	>10	>10	>10	>10	>10	>10	>10	>10
	2CS3	10 mg/mL	>10	>10	>10	>10	>10	>10	>10	>10
	2CS4	10 mg/mL	>10	>10	>10	>10	>10	>10	>10	>10
	3CS1	10 mg/mL	10	>10	10	10	>10	>10	>10	>10
	3CS2	10 mg/mL	10	>10	10	10	10	>10	>10	>10
	3CS3	10 mg/mL	>10	>10	>10	>10	10	>10	>10	>10
	3CS4	10 mg/mL	10	>10	10	10	10	>10	>10	>10
	4CS1	10 mg/mL	>10	>10	10	>10	10	>10	>10	>10
	4CS2	10 mg/mL	10	>10	10	10	>10	>10	>10	>10
	4CS3	10 mg/mL	10	>10	10	>10	10	>10	>10	>10
	4CS4	10 mg/mL	10	>10	>10	>10	10	>10	>10	>10
Positive control	Ampicillin	20 mg/mL	0.07	5	<0.15	<0.15	n.t.	n.t.	n.t.	n.t.
	Vancomycin	1 mg/mL	n.t.	n.t.	0.25	0.5	0.25	5	n.t.	n.t.
	Fluconazol	1 mg/mL	n.t.	n.t.	n.t.	n.t.	n.t.	n.t.	0.06	0.06

MIC—minimal inhibitory concentration; MBC—minimal bactericidal concentration; MFC—minimal fungicidal concentration. Values in mg/mL. S1—Sericin from Castelo Branco; S2—Sericin from Bragança; S3—Sericin purchased from Sigma-Aldrich; S4—Sericin purchased from FUJIFILM Wako Chemicals; C—*Chelidonium majus* L. 1CS—Ratio 1 *C. majus*: 1 Sericin; 2CS—Ratio 1 *C. majus*: 2 Sericin; 3CS—Ratio 2 *C. majus*: 1 Sericin; 4CS—Ratio 2 *C. majus*: 2 Sericin. Positive controls: Ampicillin, vancomycin, and fluconazol. Negative control 5% DMSO did not interfere with microbial growth. n.t.—not tested.

**Table 3 ijms-26-09911-t003:** Cytocompatibility of the different samples studied.

	C	S1	S2	S3	S4	1CS1	1CS2	1CS3	1CS4	2CS1	2CS2	2CS3	2CS4	3CS1	3CS2	3CS3	3CS4	4CS1	4CS2	4CS3	4CS4
HFF1	>400									150 ± 3.6	250 ± 8.9	>400			155 ± 4.1	333 ± 3.4	>400			182 ± 2.1	377 ± 4.7

Results were expressed as the mean values ± standard deviation. S1—Sericin from Castelo Branco; S2—Sericin from Bragança; S3—Sericin purchased from Sigma-Aldrich; S4—Sericin purchased from FUJIFILM Wako Chemicals; C—*Chelidonium majus* L. 1CS—Ratio 1 *C. majus*: 1 Sericin; 2CS—Ratio 1 *C. majus*: 2 Sericin; 3CS—Ratio 2 *C. majus*: 1 Sericin; 4CS—Ratio 2 *C. majus*: 2 Sericin.

**Table 4 ijms-26-09911-t004:** Coding information of the combined samples.

Codes	Ratio	Concentration *C. majus*/Sericin (µg/mL)	Type of Sericin (Location/Provider)
1CS1	1:1	200:1600	Castelo Branco
1CS2			Bragança
1CS3			Sigma-Aldrich
1CS4			FUJIFILM Wako Chemicals
2CS1	1:2	200:3200	Castelo Branco
2CS2			Bragança
2CS3			Sigma-Aldrich
2CS4			FUJIFILM Wako Chemicals
3CS1	2:1	400:1600	Castelo Branco
3CS2			Bragança
3CS3			Sigma-Aldrich
3CS4			FUJIFILM Wako Chemicals
4CS1	2:2	400:3200	Castelo Branco
4CS2			Bragança
4CS3			Sigma-Aldrich
4CS4			FUJIFILM Wako Chemicals

## Data Availability

The original contributions presented in this study are included in the article/Appendix A. Further inquiries can be directed to the corresponding author(s).

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
