# Peer review of "Analysis of the Phenolic Profile of Chelidonium majus L. and Its Combination with Sericin: Balancing Antimicrobial Activity and Cytocompatibility"

_ijms, 2025, doi:10.3390/ijms26209911_

Round 1

Reviewer 1 Report

Comments and Suggestions for Authors

Dear Authors,

Please find listed below my recommendations for "Analysis of Phenolic Compounds in Chelidonium majus L. and its Interaction with Sericin: A Study on Bioactive Properties." manuscript.

  1. Please ensure reliability of the citation - I observe that there are same citation that appear for multiple time and are attributed for different pharmaceuticals claims. This raise doubt about reliability. Please see Hilal et al., 2024; Maji & Banerji, 2015. Also the authors should ensure to follow the journal guidelines through manuscript editing. However in case of mentioned citation there is not clear if the cited studies represent diverse populations and extraction methods. Also the authors must evidence/refer at the dosage, bioavailability data, or clinical trial outcomes
  2. L55: This statement "broad spectrum of pharmacological activities" in my opinion should be specific and indicate which activities and their clinical relevance
  3. L57-60: What understand the authors through "substantial concentrations" - such statement should be sustained through examples/numerical values or ranges. Also in L59 the authors mention "various acid derivatives" - which specific compounds?
  4. L61-67: Here, in my opinion the authors should provide clear evidence for the synergistic interactions between compounds "synergistically contribute"
  5. L68-86: Why phenolic compounds specifically, when alkaloids (L58) were mentioned as prominent constituents? Please improve argumentation and provide proof to sustain them. L74: In my opinion such expression should not be used in a scientific paper (lauded for their potent antioxidant properties - for me this is a subjective terminology which is totally inappropriate for scientific writing). The authors refer at NF-κB pathway (L79) without context of tissue specificity or concentration-dependent effects. Please ensure accuracy and scientific relevance 
  6. L87-96; In my opinion, the authors should improve the connection between phenolic compound challenges and sericin as a solution
  7. L97-105: The authors should establish clear mechanistic rationale for sericin-phenolic compound interactions. The authors should improve the argumentation by providing precedent for similar protein-plant extract combinations. I recommend to them to address potential compatibility issues or interaction mechanisms
  8. The authors did not address enough well critical research gaps as the known hepatotoxicity concerns with C. majus - this should mentioned in my opinion. Also the gap between in vitro/in vivo studies and human clinical applications are not mentioned/ discussed. The authors should consider also in the introduction section to better discuss the translational potential. Please consider comparison with other natural antioxidant delivery systems also
  9. L134: In my opinion statements as "Natur store in Setúbal" provides insufficient traceability for scientific reproducibility. This not provide specification about collection time, plant age, or phenological stage, nor about qualified identification. The authors should consider to provide information whether roots, aerial parts, leaves, or whole plant was used//if fresh or dried material were used//what was the storage duration or conditions before extraction, etc. These kind of data should be very carefully treated by authors and provide accuracy to ensure relevance for obtained data in scientific context
  10. L137: "ethanol and water (80:20, v/v)" - what polarity was considered/provide rationale for targeting specific phytochemical classes. Please ensure clarity in statements: what was the used stirring speed/type? In case of "paper filter" (L138) the authors should mention pore size, type, and potential compound retention. It was ok that the second extraction was performed under "same conditions" for residual compounds? From my experience this often not works well
  11. L141: Considering this temperature (40 C) does the authors considered in anyway the thermolabile compounds? What understand the authors through "Reduced pressure"? quantitative parameters are mandatory in "methods" section of a research article
  12. L145: This is not ok from scientific accuracy and consistence point of view: multiple cocoon sources (2019 vs. 2023) with different storage times - this is a critical flaw; the batch heterogeneity is compromised as no standardization between different geographical sources was ensured by authors; 
  13. L155-156: Please explain this sentence! 
  14. Please provide information about quality assurance, methods performance, etc
  15. L226: The authors should very seriously consider this section as it will ensure the reliability of their presented data
  16. The authors should verify the accuracy of data reported in tables. Some compounds show precise values while others show rounded integers. There is also an inconsistent decimal place reporting across compounds. Please better evidence the replicates
  17. No relevant statistical metrics provided for data. This is a serious mistakes that should not be allowed
  18. Please better evidence  the minimum inhibitory concentrations; Improve the quantitative antimicrobial activity measurements report mode; Include  organism-specific activity profiles; Present positive or negative control results;
  19. The authors should improve their results interpretation by: including comparison with similar studies or established benchmarks; explanation of structure-activity relationships; discussion of therapeutic implications or dosing considerations; establishing correlation between chemical composition and biological activities; discuss the potential compound interactions; Please mention the compound stability or degradation potential. Please discuss the practical application challenges
  20. In this section did not placed well their findings within broader phytochemical research context. This should be improved. I strongly advice the authors to seriously reconsider their manuscript results and discussions section
  21. L452-454: This statement "effectively integrated into advanced biomaterials" is not supported by presented data. "enhance therapeutic efficacy" there are no evident clinical or in vivo evidence data provided
  22. L454-460: This statement "multifunctional wound-healing potential" is broad and without comprehensive validation
  23. The authors several times directly attribute the bioactivity to specific compounds without mechanistic studies. There are no  clear evidence of synergism between Sericin and C. majus provided. Actually the authors  present am oversimplified connections between chemical structure and biological activity which suggest a poor understanding of structure-activity relationships. Please reconsider and improve the conclusions section also
Comments on the Quality of English Language

The manuscript should better consider the language accuracy

Author Response

Dear reviewer 1,

Thank you for your thorough and insightful review of the manuscript titled "Analysis of Phenolic Compounds in Chelidonium majus L. and its Interaction with Sericin: A Study on Bioactive Properties." I appreciate the time and effort you dedicated to evaluating our work. Based on your feedback, we have made several revisions to address the issues you raised. Please see the attachment (Page 1-4).

We believe these revisions address your concerns and enhance the overall quality of the manuscript. Thank you once again for your valuable feedback and for helping us improve our work.

Best regards, Ana Borges.

Reviewer 2 Report

Comments and Suggestions for Authors

This manuscript analyzes the phenolic composition of Chelidonium majus and explores its interaction with Sericin for wound-healing applications. A total of 57 phenolic compounds were identified, and antimicrobial and cytocompatibility assays were performed across different ratios. The idea of combining C. majus phenolics with Sericin is innovative and relevant for sustainable biomaterials. However, several areas require substantial revision, particularly in mechanistic interpretation, figure clarity, and methodological reproducibility, before the paper is suitable for publication.

Comments for the authors

Comment 1. Revise the title to emphasize novelty

Comment 2. Expand the introduction with recent (2023–2024) literature on plant-derived phenolics in wound biomaterials. Integrate evidence that directly connects phenolic redox biology with tissue regeneration.

Comment 3. Integrate the most recent literature (2024–2025) on phenolic compound applications in biomedical scaffolds in the introduction section.

Comment 4. Add discussion of recent mechanistic studies showing ROS-mediated modulation by plant phenolics in biomedical contexts. A recent review on oxidative stress–driven therapeutic strategies is particularly relevant and should be incorporated here (see: 10.3390/ijms26062678).

Comment 5. The identification of 57 phenolics is impressive, but many were only tentatively assigned. Discuss the limitations of tentative MS/MS annotation without full validation and how this affects interpretation.

Comment 6. Section 3.1.1 mentions isorhamnetin without any references to back their claims. A comprehensive discussion is needed about isorhamnetin, as this carries significant biomedical relevance (see: 10.3390/ijms26157381).

Comment 7. Antimicrobial results are reported as MIC/MBC values, but there is no comparison with standard antibiotics beyond mentioning ampicillin/vancomycin. I recommend presenting fold-differences relative to controls to contextualize the bioactivity strength.

Comment 8. Discuss whether Sericin may be binding phenolics or altering their bioavailability.

Comment 9. Expand mechanistic discussion on why Sericin reduces antimicrobial efficacy, include evidence of possible binding, masking, or reduced solubility of phenolics.

Comment 10. Provide discussion on variability across Sericin sources (natural vs. commercial) and link to amino acid composition or extraction differences.

Comment 11. Compare findings with other biopolymer–plant systems in the discussion sections (e.g., chitosan–polyphenol combinations) to highlight relative novelty and translational potential.

Comment 12. Report extract yield, Sericin protein composition, and positive controls used in antimicrobial assays. Specify whether IC50/GI50 values were calculated using nonlinear regression. Provide sufficient detail for reproducibility.

Comment 13. Edit the manuscript for grammar and scientific tone. Correct tense inconsistencies, remove redundancy, and refine word choice for precision. Professional language editing is required.

Comment 14. One major issue is that the entire manuscript cites references by the first author's name, but the list provided only shows numbers. This makes it very difficult to find relevant references for the claims. The authors need to ensure consistency.

Author Response

Dear Reviewer 2,

Thank you for your thorough and insightful review of the manuscript titled " Analysis of Phenolic Compounds in Chelidonium majus L. and its Interaction with Sericin: A Study on Bioactive Properties." I appreciate the time and effort you dedicated to evaluating our work. Based on your feedback, we have made several revisions to address the issues you raised. Please see the attachment (Pages 5 and 6). 

We believe these revisions address your concerns and enhance the overall quality of the manuscript. Thank you once again for your valuable feedback and for helping us improve our work.

Best regards, Ana Borges.

Reviewer 3 Report

Comments and Suggestions for Authors

The authors demonstrate a profiling of the phenolic compounds from a C. Majus extract.Then investigate different combinations of the c.majus extract and sericin protein on different bacteria and candida by conducting an antimicrobial study and Cytocompatibility Evaluation.

The finding shows no enhancement of the antimicrobial activity of the Sericin protein and C. Majus extract relative to C. Majus extract alone.

Comment 1-

Row 245- Proteins are not Primary metabolites. Need to erase the word “proteins”

Comment 2-

Row 257- what the author means by “amino acid composition” and how can extraction methods change the “amino acids composition”?

Comment 3-

This comment refers to chapter 3.1.1 “phenolic compound profile in C. majus

The chemical profiling is valuable and provides a detailed quantification of individual phenolic compounds in C. majus, including some hydroxycinnamic acid amides not always emphasized in earlier work. While the results are largely confirmatory of previous studies, the expanded quantitative detail strengthens existing knowledge. That said, since the antimicrobial activity was tested on the whole extract rather than on the 57 identified metabolites, I would have expected untargeted profiling to capture additional compounds that may contribute to the observed bioactivity. I would suggest that the authors include the calibration curves for all metabolites that underwent absolute quantification.

Comment 4

This comment refers to chapter 3.2 “Antimicrobial activity”

The antimicrobial evaluation shows that C. majus extract alone exhibits modest but measurable bactericidal activity (MIC 5–10 mg/mL, MBC/MIC ratio 1–2). In contrast, Sericin alone was inactive, and combinations with Sericin reduced or abolished activity.

The current data also do not allow one to determine which metabolites are primarily responsible for the antimicrobial activity. No direct link has been established. Moreover, the term correlation should be avoided here unless supported by statistical analysis — it would be more accurate to describe this as a possible association. A clearer mechanistic discussion would strengthen the interpretation.

These findings contradict the proposed synergistic effect highlighted in the introduction that Sericin acts antagonistically under the tested conditions.

Comment 5-

This comment refers to the discussion:

The opening discussion sentence overstates the findings. The introduction set up Sericin as a synergistic enhancer of C. majus, yet the antimicrobial results clearly showed the opposite: C. majus extract alone was the most active, while Sericin alone was inactive and the combinations generally diminished or abolished activity. more than that, the authors claim that also later in the discussion. Therefore, statements highlighting “promising potential” or “enhanced efficacy” of the combinations are not supported by the data. In addition, the claim that activity is “closely linked” to the phenolic profile and specific metabolites such as salicylic acid, feruloyltyramine, pinocembrin, and phenylacetic acid is speculative, as no direct evidence was presented to link these compounds to the observed effects.

Rows 473-475

This paragraph is more consistent with the results than the earlier discussion sections. It correctly notes that C. majus extract alone showed the measurable antimicrobial activity (MIC 5–10 mg/mL), and that combining with Sericin generally reduced this effect in a ratio-dependent manner.

Rows 477-483

This paragraph is reasonable in noting the ratio-dependent outcomes — the 2:1 extract-to-Sericin combination retained activity while the 1:2 ratio lost it. However, the interpretation use here is overstatement. The data only show that higher C. majus content preserves antimicrobial effects, while Sericin-rich formulations abolish them. Phrasing such as “significant antimicrobial effects” and “essential role” of phenolic constituents suggests mechanistic certainty that was not experimentally demonstrated.

Author Response

Dear Reviewer 3,

Thank you for your review of the manuscript titled " Analysis of Phenolic Compounds in Chelidonium majus L. and its Interaction with Sericin: A Study on Bioactive Properties." I appreciate the time and effort you dedicated to review our work. Based on your feedback, we have made several revisions to address the issues you raised. Please see the attachment (Pages 7 and 8).

We believe these revisions address your concerns and enhance the overall quality of the manuscript. Thank you once again for your valuable feedback and for helping us improve our work.

Best regards, Ana Borges.

Round 2

Reviewer 1 Report

Comments and Suggestions for Authors

Dear Authors,

Thank you for improving your manuscript. Reading carefully the new version of "Analysis of Phenolic Compounds in Chelidonium majus L. and its Interaction with Sericin: A Study on Bioactive Properties." I have noticed several things that in my opinion should be considered. Please find them listed below:

L195-196: For me the statement as quantified using standard curves of compounds with similar chemical structure is a semi-quantitative approach at best, not true quantification. Please reconsider

Please better evidence if possible the positive control data for reference antibiotics

In my opinion the manuscript could be improved in terms of interpretation. Therefore, please better consider to evidence the answers for the following questions also: which specific compounds contribute to antimicrobial activity? how does Sericin interact with phenolic compounds? what is the mechanism of antimicrobial action? and/or what cellular pathways are affected?

Author Response

Dear reviewer 1,

Thank you for taking the time to provide a second thorough and insightful review of our manuscript, originally titled “Analysis of Phenolic Compounds in Chelidonium majus L. and its Interaction with Sericin: A Study on Bioactive Properties.” We sincerely appreciate the effort and careful consideration you dedicated to evaluating our work.  Please see the attachment (Page 1).

We believe these revisions address your concerns and enhance the overall quality of the manuscript. Thank you once again for your valuable feedback and for helping us improve our work.

Best regards, Ana Borges.

Reviewer 2 Report

Comments and Suggestions for Authors

I recommend accepting the paper in its present form.

Author Response

Dear Reviewer #2,

Thank you very much for your thorough and insightful review of our manuscript, initially titled "Analysis of Phenolic Compounds in Chelidonium majus L. and its Interaction with Sericin: A Study on Bioactive Properties." We greatly appreciate the time and effort you dedicated to evaluating our work.

  1. I recommend accepting the paper in its present form.”: We are particularly grateful for your positive recommendation. Your encouraging feedback confirms that the revisions carried out have strengthened the manuscript and improved its clarity, accuracy, and scientific relevance.

Once again, we sincerely thank you for your constructive comments and support throughout the review process.

Best regards, Ana Borges.

Reviewer 3 Report

Comments and Suggestions for Authors

Review comments

Comment 1- rows 259-266

The authors suggest that Sericin could interact with polyphenolic compounds through hydrogen bonding and electrostatic interactions, given its high content of serine, aspartic acid, and glycine. This is plausible and, “on paper,” could be true. however, serine, aspartic acid, and glycine cannot engage in π–π stacking, as they do not possess aromatic rings.

Comment 2- rows 267-272

While the combined use of Sericin and polyphenols in wound-healing biomaterials is a reasonable direction, the authors currently assumes direct protein–polyphenol interactions without providing supporting evidence. In addition, the statement that “protein–polyphenol interactions are likely to influence stability and bioactivity” is vague. It is unclear whether “stability” refers to polyphenol chemical stability, stability of the complex, or the biomaterial matrix. Moreover, the stability of protein–polyphenol complexes is not necessarily advantageous for bioactivity, as tight binding could hinder the release of polyphenol metabolites and thus reduce their functional activity.

Comment 3-

The introduction emphasizes the potential synergy of combining Sericin with C. majus, especially with respect to antimicrobial activity, creating the expectation that Sericin will potentiate or at least preserve the antimicrobial effects of C. majus. however, the experimental results clearly demonstrate the opposite: Sericin consistently reduced or abolished antimicrobial activity and the authors do acknowledge it in the results and in the discussion . while the authors mention ratio-dependent pattern in the discussion, this effect is entirely negative, and its “concentration depended” in did but not in a positive or balance way, with only C. majus-rich formulations retaining partial activity. these findings represent “negative results.” While negative results can be valuable, the manuscript does not sufficiently explain their significance or how they advance the field. I encourage the authors to explicitly address the apparent contradiction between the introduction and the results, and to clarify what new insight is gained from showing that Sericin attenuates, rather than enhances, the antimicrobial effects of C. majus.

Comment 4-

The current title suggests that the study demonstrates “bioactive interactions” between C. majus phenolic compounds and Sericin, implying positive or synergistic effects. but the results do not actually show any direct interactions between individual metabolites, phenolic compounds, and Sericin, nor do they demonstrate enhanced bioactivity from the combination. On the contrary, the antimicrobial activity of C. majus is attenuated when Sericin is added. I recommend revising the title to avoid overstating the findings and to more accurately reflect the data

Author Response

Dear reviewer 3,

Thank you for taking the time to provide a second thorough and insightful review of our manuscript, originally titled “Analysis of Phenolic Compounds in Chelidonium majus L. and its Interaction with Sericin: A Study on Bioactive Properties.” We sincerely appreciate the effort and careful consideration you dedicated to evaluating our work.  Please see the attachment (Pages 3 and 4).

We believe these revisions address your concerns and enhance the overall quality of the manuscript. Thank you once again for your valuable feedback and for helping us improve our work.

Best regards, Ana Borges.
